# Knowledge-Based Decision Support in Healthcare via Near Field Communication

**DOI:** 10.3390/s20174923

**Published:** 2020-08-31

**Authors:** Giuseppe Loseto, Floriano Scioscia, Michele Ruta, Filippo Gramegna, Saverio Ieva, Agnese Pinto, Crescenzio Scioscia

**Affiliations:** 1Department of Electrical and Information Engineering (DEI), Polytechnic University of Bari, via E. Orabona 4, 70125 Bari, Italy; giuseppe.loseto@poliba.it (G.L.); floriano.scioscia@poliba.it (F.S.); filippo.gramegna@poliba.it (F.G.); saverio.ieva@poliba.it (S.I.); agnese.pinto@poliba.it (A.P.); 2Department of Emergency and Organ Transplantation (DETO) Rheumatology Unit, University of Bari, Piazza G. Cesare 11, 70124 Bari, Italy; crescenzio.scioscia@policlinico.ba.it

**Keywords:** decision support, ubiquitous healthcare, knowledge graph, automated reasoning, near-field communication

## Abstract

The benefits of automatic identification technologies in healthcare have been largely recognized. Nevertheless, unlocking their potential to support the most knowledge-intensive medical tasks requires to go beyond mere item identification. This paper presents an innovative Decision Support System (DSS), based on a semantic enhancement of Near Field Communication (NFC) standard. Annotated descriptions of medications and patient’s case history are stored in NFC transponders and used to help caregivers providing the right therapy. The proposed framework includes a lightweight reasoning engine to infer possible incompatibilities in treatment, suggesting substitute therapies. A working prototype is presented in a rheumatology case study and preliminary performance tests are reported. The approach is independent from back-end infrastructures. The proposed DSS framework is validated in a limited but realistic case study, and performance evaluation of the prototype supports its practical feasibility. Automated reasoning on knowledge fragments extracted via NFC enables effective decision support not only in hospital centers, but also in pervasive IoT-based healthcare contexts such as first aid, ambulance transport, rehabilitation facilities and home care.

## 1. Introduction

Automatic identification (AutoID) technologies rely on information stored on transponders (tags) and retrieved by interrogator devices (readers) through contactless short-range radio signals. Widespread AutoID technologies include Radio Frequency IDentification (RFID) [1] and Near Field Communication (NFC) [2]. Tags can be applied to or incorporated into objects, animals, or people for identification and tracking. They typically contain: (i) either a unique code, which is read by the interrogator and used to identify the associated object/subject or (ii) a Uniform Resource Identifier (URI) to trigger interactions with Web applications/services. Technological progress in transponder miniaturization, power consumption and memory availability opens new interesting implementation possibilities [3]. If tagged items could expose an articulated annotation to readers instead of a simple numeric identifier, they might self-describe without depending on a centralized information server [4]. This would be particularly useful in case: (i) a stable network connection toward the remote information repository is not available; (ii) accessing information regarding the object/subject requires maximal availability and minimal latency; (iii) data stored within the tag has to provide an expressive representation of object/subject features and capabilities to enable knowledge inference.

All the above situations are specifically relevant in ubiquitous healthcare (u-healthcare), which includes applications and services both for clinical facilities and for tele-medicine, tele-rehabilitation, and tele-homecare. In fact, AutoID technologies are increasingly adopted in healthcare management systems as well as (self-)diagnosis, (self-)medication and assistive solutions [2]. Within hospital premises, current benefits mainly consist in error prevention in tracking equipment, identifying staff, and regulating access to various locations for patients and practitioners [5]. A more advanced exploitation of mobile and pervasive computing technologies like NFC—supported by innovative Human–Computer Interaction (HCI) paradigms— could further enhance their impact in e-healthcare, but there is no systematic framework yet for incorporating them.

This paper presents a ubiquitous Decision Support System (DSS) for innovative healthcare solutions, based on a semantic enhancement of NFC standards. The proposed framework helps physicians validate, confirm and choose the best personalized therapy for the patient’s medical record. The NFC Data Exchange Format (NDEF) [6] of the NFC Forum has been extended to enable support for knowledge representation. The Web Ontology Language (OWL) version 2 [7] by the World Wide Web Consortium (W3C) is the adopted knowledge formalism. Any enhancement ensures compatibility with legacy NFC applications. NFC tags attached to, e.g., pharmaceuticals packaging and patients’ NFC wristbands become able to store semantic metadata, which are exploited in a matchmaking process evidencing possible problems in a given therapy with respect to the clinical profile of the patient, as well as recommending further personalized treatment options to the healthcare practitioner.

The proposed framework complies with Cyber-Physical System (CPS) precepts in pervasive Internet of Things (IoT) contexts. A background knowledge model, exploiting and expanding a given medical Knowledge Graph (KG), provides the needed terminology for non-standard deductions [8]. By exploiting NFC and mobile computing devices, straightforward HCI patterns are applicable to a range of u-healthcare contexts. They include not only hospital centers, but also rehabilitation facilities, homecare and even fully mobile scenarios such as first aid and ambulance transport. Main contributions of the proposed DSS framework and system prototype concern:A mobile and pervasive IoT architecture for context-aware decision support;A knowledge model for patient case history, diseases and treatments congenial to automated inference and extensible to every medical branch;Adaptation of NFC to store semantic descriptions of patient case history and medications;Non-standard inferences set for semantic matchmaking in healthcare;A mobile user interface that provides context-aware automatic decision support during normal therapy management workflows with visual suggestions and cues.

Both NFC and RFID AutoID technologies could be adopted theoretically in the proposed DSS. NFC has been preferred by virtue of its ability to protect sensitive healthcare information from eavesdropping and man-in-the-middle attacks—due to short radio range—and to support bidirectional interactions [9]. Furthermore, NFC technology is universally standardized at the 13.56 MHz UHF (Ultra-High Frequency) band, while several RFID standards exist for specific classes of use cases, adopting different protocols and frequency bands. This has facilitated the integration of NFC reader functionalities in a significant portion of currently available mobile devices (smartphones, tablets), which are immediately compatible with the proposed DSS. Conversely, RFID reader peripherals or dedicated devices should be purchased to enable the proposed vision. Moreover, all NFC tags are passive, i.e., they need no power source to work: this helps drive costs down with respect to active RFID tags. The framework architecture and HCI workflows have been designed to mitigate the main challenges of NFC, which make it unsuitable to other scenarios:The very short communication range of NFC becomes a benefit rather than an issue in this case, due to the above security reasons.Data compression techniques integrated in the framework cope with the limited data storage space of NFC tags.The maximum data transfer rate of NFC is 424 kbit/s, slower than other wireless protocols. Nevertheless, the reduced size of annotations allows a rapid data exchange between applications and NFC tags.One NFC tag can be scanned at a time, whereas RFID protocols allow multiple tags to be scanned simultaneously. This is not a problem, however, since healthcare professionals exploiting the framework always interact with one patient at a time.NFC is unfitting for asset tracking applications: RFID remains a better choice for that kind of scenarios in healthcare facilities, but the proposal in this paper does not need item tracking.

The remaining of the paper is organized as in what follows. The next section discusses relevant related work on NFC and DSSs in the healthcare field. Section 3 describes the proposed approach, outlining the system architecture and decision support framework. Section 4 illustrates the system prototype and clarifies benefits of the proposal in two example scenarios extracted from a rheumatology case study. Finally, Section 5 reports on system evaluation before conclusion.

## 2. Related Work

In healthcare, different types of practitioners interact with patients in various activities, which are typically coordinated in complex workflows, possibly involving several territorial Point-of-Care (POC) facilities [5,10]. Capabilities of NFC technology [9] can be leveraged to improve healthcare operations. Within a hospital, readers can be deployed at key points of passage and locations to track staff, equipment and patients tagged with NFC transponders. Actions triggered or recognized via smart devices can be logged automatically, thus avoiding lengthy and error-prone manual data input by personnel. Research studies and pilot projects have pointed out critical design and implementation issues [11,12] and have evaluated the impact of such infrastructures in ordinary hospital activities [13] as well as in Intensive Care Units (ICUs) [14]. This increases security and safety by automating checks for authorization enforcement and prevention of human error, respectively, in critical activities like medication administration [15]. Furthermore, medicine counterfeiting can be prevented [16] and NFC-enabled wearable biosensors have been proposed [17,18]. So far, however, NFC has not been exploited in more knowledge-intensive healthcare tasks, such as Clinical Decision Support (CDS).

The integration of radio-frequency devices with other pervasive computing technologies—such as communications protocols and wireless sensor networks—is leading to further innovative applications in the tele-medicine area [19], particularly for ubiquitous persistent monitoring [20,21] Context-awareness is the key aspect of such approaches to improve quality of healthcare services. Challenges and benefits have been clearly evidenced since the earliest RFID-based HMS (Healthcare Management System) prototypes [22]. The HMS in [23] exploited NFC tags to store patients’ basic data created during the first hospital admission. On the other hand, patient’s clinical history was stored only in the hospital database and the doctor obtained access during visit or medication prescription by using the patient’s NFC tag as a key; NFC was used to authorize wearable sensors to store tele-homecare measurements in the hospital database as well. Similarly, in [24] NFC tags attached to hospital beds stored unique hashes for accessing patients’ electronic health records in the hospital information system and ensuring proper staff authentication and authorization. A smartphone app assisted nurses in medication delivery, but it only allowed checking delivered medications off the patient’s prescription list; no medication appropriateness check could be performed. Likewise, the bedside medication administration system in [15] exploited NFC armbands to check the so-called “five rights" (giving the right medication to the right patient in the right dose by the right route at the right time) with lower error rates than other methods, thanks to context-aware nurse interaction, but it did not facilitate checking medication interactions and contraindications: this remained a challenging task for nurses, as it required pharmacological knowledge. Another security-focused IoT HMS is in [25], designed to provide decision support in remote health monitoring. The system performed anomaly detection through Support Vector Machines (SVM) machine learning technique; correlation between pairs of sensor data streams was exploited to cope with missing values and to provide decision support in case of anomaly. The latter, however, relied on “medical rules” which were assumed but neither formalized nor integrated in the proposed framework: that limitation highlights the need for knowledge-based CDS like the one proposed in this paper. All the above systems, in fact, provided only basic identification features and lacked more advanced knowledge-based capabilities.

Computerized CDS is acknowledged among the most significant benefits of medical informatics [26]. Specifically, research suggests rule-based and artificial intelligence systems can help health practitioners avoid diagnosis and treatment errors [27]. Nevertheless, the first generation of Computerized Physician Order Entry (CPOE) systems was mostly based on manual data entry and a fragmented collection of non-integrated utilities. Experience, however, has taught the overall effectiveness of CDS depends on multifaceted context, system and implementation factors [28], as “decision support is highly effective only when it is automatic and seamless” [26]. A recent systematic review [29] has found three factors making CDS more effective: automatic provision of decision support, on-screen display of advice and patient-specific suggestions. The requirement of greater levels of personalization in CDS is in accordance with findings that the adoption of computerized medical systems appears to yield poorer outcomes for atypical cases [30].

Semantic-based technologies are a key enabler of personalized and adaptive systems [31]. In the healthcare sector, they can be leveraged to ensure that physicians get highly significant information about the clinical status of a patient and the most appropriate treatments for the specific case, thus supporting unobtrusive and context-aware decision support services for therapy management. So far, semantic-based frameworks exploiting NFC have been mostly focused on capturing context-awareness information [32] or personalizing application functionality [33]. In these works, however, NFC was used just as a HCI facilitator: it conveyed merely low-level data either used to trigger semantic-enabled information processing or collected to be analyzed through ontology-based models. No information with explicit semantics was manageable on transponders. In [34] NFC wristbands were used to store patients’ health data which could be read or updated by the various practitioners in a hospital facility. While data mobility enabled designing clinical processes as workflows in a multi-agent system architecture, the lack of formal and structured information representation prevented automating knowledge-intensive tasks. On the contrary, the early RFID-based system in [22] provided decision support capabilities in therapy assignment, although relying only on exact matches between the list of adverse effects in a medicine record and the list of diseases and allergies in a patient’s record, both retrieved from a centralized database through Structured Query Language (SQL)-like queries. Logic-based matchmaking enables detecting also risks that are not explicit, but are consequences of modeled knowledge concerning pharmaceuticals and diseases.

The use of widely-adopted KR languages and technologies—borrowed from the Semantic Web initiative and adapted for efficiency in resource-constrained mobile contexts—can promote interoperability and integration of solutions designed for hospital centers hosting tele-medicine applications: an example can be found in [35], presenting an Ambient-Assisted Living (AAL) IoT platform exploiting the oneM2M standard [36] for semantic interoperability in distributed systems. Physical objects and devices are represented by virtual objects (VOs), annotated with respect to the oneM2M Base Ontology (available at https://git.onem2m.org/MAS/BaseOntology) in OWL language, serialized in JSON (JavaScript Object Notation) or XML (Extensible Markup Language) syntax. VOs can be discovered by means of SPARQL [37] queries and composed through a semi-automatic tool, exploiting a customized OWL-S [38] (Semantic Markup for Web Services) ontology. Unfortunately, the proposal only supports full match in VO discovery and composition: the lack of approximate match support limits recall—and thus, usefulness—in practical contexts, where objects are highly heterogeneous and annotations should be as detailed as possible to enhance the precision of retrieval.

As pointed out in [39], pervasive computing technologies allow gathering health data at an unprecedented scale: knowledge-based approaches can assist in the management, analysis and interpretation of such data for research purposes and/or to improve clinical best practices. The framework presented in [40] and updated in [41] took a first step in this direction, by combining a semantic-based enhancement of EPCglobal RFID protocol with healthcare decision support features based on Knowledge Representation (KR) technologies. This paper updates the above frameworks from several viewpoints: (i) technological, by integrating currently pervasive mobile devices and NFC communication; (ii) algorithmic, by expanding the inference capabilities and adding new decision support features; (iii) standardization, by adopting OWL instead of DIG (Description Logic Implementation Group) language as well as providing a clearer and more complete specification of the proposed Knowledge Graph to support inferences, also referencing well-known KGs in the medical domain as per Linked Data best practices [42].

Pervasive computing technologies are prevalent in decision support systems for remote health monitoring and tele-homecare, which rely on methods for AAL and Human Activity Recognition (HAR). Differently from the above proposal [25] relying on machine learning, the AAL DSS in [43] adopts a knowledge-based approach, using semantic representations of dwellers and domestic environments to recommend the adoption of home appliances able to assist elderly people in coping with their limitations and living an independent life. Like in this paper, the International Classification of Functioning, Disability and Health (ICF) framework of the World Health Organization (WHO) [44] is exploited to characterize individual impairment levels. Dweller and appliance profiles are annotated with reference to an OWL ontology. The discovery of appliances suitable to help dwellers occurs through SPARQL queries based on Semantic Web Rule Language (SWRL) [45] rules: unfortunately, the system can retrieve full matches only, while the approach proposed in this paper ranks approximate matches as well, by means of non-standard inferences, when retrieving treatment options.

Machine learning (ML) is increasingly adopted for HAR. In [46] nine well-known ML techniques were compared in a HAR problem with two different frameworks, based on wearable sensors and smartphone sensors respectively. SVN provided the best results in both tests; unfortunately, it is a black box technique, i.e., the trained model is not transparent and the outcome is a trivial classification label, without interpretable information annotation. This affects other machine learning approaches as well [47], preventing both outcome explainability and integration with knowledge-based systems like the one proposed in this paper. A semantic-based posture and gesture recognition system can be found in [48], based on popular three-dimensional (3D) motion sensors like Asus Xtion Pro Live: sequences of skeleton representations returned by the sensing device are annotated with respect to an OWL ontology, so that it undergoes semantic matchmaking with a knowledge base of attitude and gesture templates to retrieve the best matching ones.

Table 1 summarizes the comparison of the proposed approach with related systems integrating identification and/or tracking technologies. With the exception of [41,48], other works provide only exact query support or machine learning black-box algorithms, lacking ranking and explanation capabilities. While this work is an upgrade of [41] in several aspects, ref. [48] has some similarities as a knowledge-based system, but the underlying sensing technologies (Kinect vs. NFC) and use cases (HAR vs. CDS) are different. Finally, it is useful to remark how this work, refs. [35,43] demonstrate the benefits of Linked Data by leveraging and expanding well-known ontologies.

## 3. Proposed Approach

In the following subsections, both architecture details and the algorithms supporting the DSS inferences are presented.

### 3.1. Architecture

As depicted in Figure 1, the proposed clinical decision support system runs on a mobile device (smartphone or tablet) equipped with: (i) NFC reader peripheral for extracting annotated descriptions stored in NFC tags attached to patients’ wristbands, caregivers’ badges, pharmaceutical packagings and medical equipment; (ii) touchscreen for interaction with the caregiver, via a Graphical User Interface (GUI); (iii) a lightweight embedded reasoning engine [8], providing an optimized implementation of standard and non-standard inference services. The above core architectural elements do not require centralized back-end infrastructures. Consequently, the proposed framework allows on-the-fly, in-place decision support to caregivers not only within hospital premises, but also in rehabilitation facilities, for (tele-)home care and in ambulances.

The groundwork for the overall infrastructure consists in a semantic enhancement of the NFC protocol. All tags compliant with the NDEF standard [6] can be used to host semantic descriptions and data-oriented contextual parameters. Through novel exploitation of the NDEF record fields, no modification is needed to standard NFC equipment and communication primitives, so allowing u-healthcare deployments to save on hardware investments. Basically, each NDEF message contains multiple records whose format is shown in Figure 2. The Type Name Format (TNF) field is a 3-bit identifier of the type and the content of a record, as reported in Table 2. The proposed approach exploits Multipurpose Internet Mail Extensions (MIME) record type (TNF = 2) to share rich and unambiguous semantic annotations about relevant healthcare entities. Web Ontology Language (OWL) 2 [7] has been adopted as reference language. As OWL provides different serialization syntaxes, the following standard MIME types can be specified in the Type field of the NDEF record to identify the right format of the annotation carried in the Payload: application/rdf+xml for RDF/XML [49]; application/owl+xml for OWL/XML [50]; text/owl-functional for functional-style syntax [51]; text/owl-manchester for Manchester syntax [52]; text/turtle for Turtle syntax [53]. A record ID is also associated to each NDEF message—as shown in Figure 2—to unambiguously identify the payload. Only the first record in a message contains the ID, whereas for subsequent ones this field will be empty. The identifier is generated according to the unique Internationalized Resource Identifier (IRI) of the reference knowledge graph (KG) and the selected task (e.g., retrieve patients, caregivers or pharmaceutical annotations). In this way, a reader is able to purposely filter given messages. Moreover, the framework supports compression algorithms (detailed in Section 5) aiming to decrease the size of OWL annotations, fitting the limited memory amounts of current NFC tags.

### 3.2. Knowledge Graph Modeling

The Entity-Relationship diagram in Figure 3 provides a high-level view of the Knowledge Graph (KG) underlying the proposed framework. Each instance of patient, caregiver and treatment is described by an annotation along with a set of quantitative attributes. Annotations are stored in individual NFC tags.

Description Logics (DLs) are a family of logical languages adopted as reference formalism in the proposed framework—the readership is assumed to be familiar with basics of DLs and is referred to [55] for examples and wider argumentation—and specifically the OWL 2 subset whose semantics maps to the ALN (Attributive Language with unqualified Number restrictions) DL. The proposed KG is composed by: (i) a formal conceptual model of the medical domain (the ontology a.k.a. Terminological Box T in DL words), shown in Figure 4 and intended as background knowledge for the DSS; (ii) assertions about instances (a.k.a. the Assertion Box), i.e., patients’ medical record and pharmaceuticals annotations, retrieved on-the-fly via NFC. In detail the KG is structured as:Ontology (T):
-Anatomy: taxonomy of body structures and systems;-Taxonomy of treatment classes, enriched with general adverse effects and interactions (either positive, negative or dangerous ones) with other treatment classes;-Taxonomy of diseases, related with affected body structures and typical treatment classes;-Healthcare professional specialties.Instances:
-Healthcare practitioner’s profile;-Patient’s medical record: general information (e.g., age), disease(s) and current treatment(s);-Treatment description, particularly treatment class and specific adverse effects.

According to Linked Data principles and best practices [42], the proposed Smart Health (SH) KG (available at http://sisinflab.poliba.it/swottools/onto/sh/) reuses and combines well-known KGs. Basically, the Systematized Nomenclature of Medicine—Clinical Terms (SNOMED-CT) [56] has been adopted as upper ontology to model the taxonomy of both human anatomy and medical specialties. Moreover, specific clinical terms have been borrowed from the Uber Anatomy Ontology (UBERON) [57] and the Foundational Model of Anatomy (FMA) [58], extending the basic anatomical taxonomy with additional entities about human systems. The Ontology for General Medical Science (OGMS) [59] has been referred for terms pertaining to diseases, diagnoses and clinical phenotypes; Human Disease Ontology Identifiers (DOID) [60] for modeling concepts for human disease representation; the Chemical Entities of Biological Interest Ontology (ChEBI) [61] for proposing a structured classification of chemical compounds useful to feature medicines.

Since the system validation has been conducted in a rheumatology ward (see Section 4), the KG includes a complete model only for connective tissue diseases, a specific class of rheumatic pathology. Other disease classes have been modeled using more generic concepts. This has allowed to control the modeling effort while being able to fully evaluate the effectiveness of the proposed knowledge-based approach for decision support, albeit in a single medical branch. By extending the KG along the adopted patterns, other diseases and therapies can be modeled in order to expand the domain knowledge to further medical fields.

Extra-logical data-oriented attributes are used to integrate and refine the automated reasoning outcomes, in order to take context-aware parameters into account in decision support. They are modeled as OWL annotation properties related to the instances of patients, caregivers and medicines. Also in this case, well-known semantic vocabularies have been exploited. For patients: name, surname, gender and weight are defined through the schema.org vocabulary [62]; age is modeled following the FOAF specification [63]; severity of condition is represented in a scale from 0 to 4, following the WHO ICF framework [44]. Figure 5 shows an example of patient model. As depicted in Figure 6, attributes for pharmaceuticals include: name, manufacturer and generic description defined by schema.org; dosage and frequency of adverse effects using the drugUnit and warning properties of the Health and Lifesciences vocabulary (https://health-lifesci.schema.org/), an extension of schema.org proposed for medical entities; Anatomical Therapeutic Chemical (ATC) identification code, based on the WHO ATC classification system [64]; AIC code (https://www.fascicolosanitario.gov.it/sistemi-codifica-dati/informazioni/aic), an authorization number issued by the Italian Drug Agency (Agenzia Italiana per il Farmaco, AIFA) for verifying the authenticity of a medicine package. The last two attributes refer to specific properties defined in the Smart Health KG as sub-properties of schema:code provided by schema.org. Finally, caregivers’ annotations include name, surname and description in terms of position or task.

### 3.3. Reasoning for Healthcare Decision Support

Non-monotonic inference procedures in [55] are leveraged to discover suitable treatments for a given patient’s disease, personalized for the individual medical history. The matchmaking engine [8] embedded in the DSS is able to compute a score based on the semantic consistency between patient’s disease(s) and available medication profiles, in order to: (i) detect possible incompatibilities between a proposed therapy and the patient’s condition; (ii) find out interactions with treatments the patient is already undergoing, which may result in lower therapy effectiveness or health risks; (iii) rank treatment options by appropriateness; (iv) explain the outcomes of matchmaking in all cases.

The proposed framework compares DL concept expressions for a patient’s medical profile *P* and a medication description *D* by means of Concept Contraction (CC) and Concept Abduction (CA) non-standard reasoning services [55]. Particularly, in case of a partial match between *D* and *P* (i.e., *D* and *P* clash in some part), the result of CC is a pair 〈G,K〉 representing what has to be given up (i.e., retracted) *G* and what can be kept *K* in *D*, respectively, in order for *K* to achieve a potential match with *P*. Basically, *G* consists in the elements of *D* conflicting with *P*, while *K* is the (best) contraction of *D* compatible with *P*. On the other hand, in case of a potential match of *D* with *P* (i.e., there are no conflicting aspects, but *P* does not completely satisfy *D*), CA returns the concept *H* representing what has to be hypothesized in *P* to reach a full match with *D* (or its contracted version *K*). Penalty scores computed on *G* and *H* are associated to Concept Contraction and Concept Abduction, respectively, and formal minimality criteria exist to give up or hypothesize as little as possible [55].

The final purpose of the proposed approach is to assess how much a given medication fits the patient’s diseases, in order to provide therapy decision support to practitioners. This is achieved by enriching the disease annotation with pharmaceutical classes appropriate to treat the disease itself. This allows Concept Abduction to check whether a given treatment is appropriate or not. Moreover, descriptions of diseases and treatments use disjoint concepts to refer to affected bodily organs and systems. In this way, if a given pharmaceutical exhibits adverse effects for the specific patient’s profile, the Concept Abduction check will fail due to semantic inconsistency and the Concept Contraction algorithm will detect what parts of a therapy annotation are the cause of the contraindication. A similar approach, based on disjoint concepts, is adopted also to detect interactions between the new treatment to be administered and previous treatments the patient is already undergoing.

Figure 7 shows the DSS workflow for therapy verification. Steps are summarized hereafter:The system executes Concept Abduction of treatment description *D* with respect to patient’s profile *P*, both extracted via NFC:
1.a.If *D* and *P* are compatible (i.e., their conjunction D⊓P is satisfiable with respect to the reference ontology T), the proposed therapy is approved;1.b.If *D* and *P* are incompatible, instead, Concept Contraction extracts the contraindications and the therapy interactions of the treatment;The patient’s profile is submitted to a matchmaking process against treatments defined in the KG in order to find a new medication $D^{\prime }$; this step yields a list of substitution therapy options sorted by relevance. Penalty functions associated to CA and CC solutions [55] are used to measure the semantic distance (hence the affinity level) between the patient’s medical profile and each medication annotation;In order to improve flexibility of decision support, the semantic distance is combined with context-specific variables through a utility function, whose details are provided in Section 4. Results are ranked according to the overall score.

## 4. Case Study

A fully functional mobile prototype has been developed using Android SDK Tools (revision 26.1.1) for the Android Platform version 8.1 (Application Programming Interface—API—level 27). Illustrative examples should clarify the proposed framework by presenting the therapy verification and decision support workflow in a rheumatology unit, with patients affected by connective tissue diseases. Let us consider two purportedly simple examples:

**Example** **1.**
*A patient named S. P. has paraneoplastic dermatomyositis (PDM) and anemia. This is expressed (in OWL 2 Manchester syntax) with respect to the DSS KG as:*

S_P: PDM **and** Anemia


A practitioner approaches his/her bed with a mobile device running the decision support system. The doctor activates the system by placing the NFC reader near his/her badge. Information extracted from the embedded tag is shown and authorization is checked. If the DSS is activated by an unauthorized person—e.g., a medical student or a physician from another ward—it blocks interaction with an alert and logs the event. In a similar way, the patient’s profile is retrieved by placing the mobile device near his/her NFC wristband and it is shown on the screen (Figure 8a). The doctor intends to administer cortisone as therapy. During therapy assignment, the DSS supports physicians by retrieving the proposed therapy annotation from the KG, while during therapy administration it assists practitioners—e.g., nurses—by checking the medicine annotation extracted from the NFC tag on the package. In either case the mobile graphical user interface (GUI) shows basic information stored in the tag as displayed in Figure 8b. In the example, the semantic annotation for cortisone includes potential adverse effects towards eyes, bones and cardiovascular system:


Cortisone: Corticosteroid and (therapy only Corticosteroid) and (affects only (Eye and Bone and CardiovascularSystem))


S. P. is affected from a blood disease, while the KG ontology specifies PDM can affect the circulatory and skeletal systems. Attention is required in the treatment of PDM with cortisone. When the doctor touches the “continue" button, the system performs this inference automatically and issues a warning along with the semantic-based affinity level between cortisone and the patient’s profile (Figure 8c). In the example, the algorithm in Section 3.3 produces the following result:


Give up: affects only (CardiovascularSystem and Bone)



Keep: Corticosteroid and (therapy only Corticosteroid) and (affects only Eye)


According to the inference outcome, the embedded matchmaker computes $penalty_{CA}$ with the patient’s description. If it detects incompatibility, it executes Concept Contraction to extract the conflicting Give-up part of the request and evaluates the induced $penalty_{CC}$ value, then it computes $penalty_{CA}$ again on the remaining Keep part. An overall score combination function (a.k.a. utility function) combines results of the matchmaking framework with the following context-specific parameters:Patient’s age;Incidence rate of medication adverse effects;WHO ICF severity of patient’s disease-related impairment, represented in integer scale as follows: 0= no impairment, 1= mild, 2= moderate, 3= severe, 4= complete [44].

The utility function is defined as follows:$$f_{sc}=\frac{penalty_{CC}+penalty_{CA}}{max_{penalty_{CA}}}\cdot tanh\frac{age}{\alpha }\cdot severity\cdot tanh\frac{adv\_rate}{\beta }$$


Since $penalty_{CC}$ and $penalty_{CA}$ are computed as semantic distance measures, also the overall function is modeled to represent better matches with lower values. The first factor measures the compatibility of medications with the patient’s medical profile: in particular, $max_{penalty_{CA}}$ is the highest Concept Abduction penalty among all medications in the KG (i.e., the least effective treatment). The next two factors take into account the fact that, in general, patients with younger age or lower ICF impairment can tolerate treatments better (notice that the model does not apply to pediatric patients). Contraindications are modeled in the final factor, with adv_rate the number of occurrences of adverse events per 100 patients. the Weights α and β are tunable: empirical evaluation has suggested values α=50 and β=10.

The description of the conflicting characteristics is shown in the Adverse Effects tab depicted in Figure 8d, highlighting anatomical parts actually affected by the administration of the medication. The final decision is left to the doctor, who can confirm the therapy or query the DSS for additional treatment options. In the latter case, by applying the above inference task to the other medication instances in the KG, the system returns an ordered list of substitution therapies as depicted in Figure 8e, which is shown to the physician in the "Suggestions" tab. For each treatment, the GUI displays the affinity score, calculated through the above formula and converted into an ascending [0,100%] range, the incidence rate of adverse events and an explanation section showing possible adverse effects and interactions of the substitution therapy. In our running example, adalimumab differs from cortisone as it is an Anti TNFα, but it is equally suitable for PDM. Also in this case the system identifies its risks related with the medical profile of patient S. P. and displays them as in Figure 8f; however adalimumab is preferable as its adverse events incidence rate is lower (1 in 100 vs. 6 in 100). The physician selects an appropriate therapy. By scanning the NFC tag in the patient’s transponder again, the mobile device stores the updated profile including the assigned therapy.

**Example** **2.**
*The second example aims to clarify how the proposed system manages conflicts that could occur between a new therapy and the ones the patient is already undergoing. The profile for patient C. C. includes information about the medication(s) s/he already assumes, namely a beta blocker. As shown in Figure 9a, the patient is affected by a mild form of Systemic Lupus Erythematosus (SLE) and a generic disease of the muscular and skeletal system. This is expressed with the following formula:*

C_C: MildSLE and MusculoskeletalSystemDisease and (seriousEffect only BetaBlocker) and (reduces only BetaBlocker) and (reducedBy only BetaBlocker) and (increases only BetaBlocker) and (increaseBy only BetaBlocker)


A rheumatology nurse tries to give her ibuprofen (a nonsteroidal anti-inflammatory drug, NSAID). For this patient the system is able to recognize not only the adverse indications between the new medicine and the patient’s diseases, shown in Figure 9b like in the previous example, but also possible undesired therapy interactions: matchmaking of ibuprofen and the patient’s profile reveals that NSAID class reduces the effect of beta blockers. The detected interaction is therefore displayed on the "Interactions" tab, as shown in Figure 9c.

The proposed examples show that non-standard inferences on KGs can be leveraged to provide useful decision support, even though expressiveness of the logical language is limited by the need to provide acceptable reasoning performance on mobile devices.

## 5. Evaluation

Performance assessment of the proposed framework and implementation are outlined hereafter. The mobile prototype has been tested on a Samsung Galaxy S6 smartphone with Exynos 7420 quad core CPU at 1.5 GHz, 3 GB RAM, 64 GB internal memory, and Android version 7.0. The choice of a 5-year-old smartphone model has had the purpose to assess whether the proposed approach could achieve acceptable computational performance even on relatively resource-constrained mobile hardware. For this experiment, NFC tags have been emulated via the Android Beam API (https://developer.android.com/guide/topics/connectivity/nfc) on a Google LG Nexus 4 smartphone with Qualcomm APQ8064 Snapdragon S4 Pro quad core CPU at 1.5 GHz, 2 GB RAM, 16 GB internal memory, and Android version 5.1.1.

*NFC data exchange.* In order to reduce the size of annotations stored in each tag, the proposed implementation supports the following compression algorithms: (i) EXI (Efficient XML Interchange) [65], a compact representation for XML-based OWL syntaxes; (ii) HDT (Header, Dictionary, Triples) [66], proposed to compress Turtle syntax while maintaining query capability without decompression; (iii) LZMA, GZIP and Deflate, general-purpose algorithms suitable for all OWL syntaxes and provided by the Apache Commons Compress library (http://commons.apache.org/proper/commons-compress/). Selected algorithms were tested on 19 OWL 2 annotations corresponding to different patients, caregivers and treatments profiles. Table 3 reports on the average size of the annotations with and without compression. Deflate provided the overall shortest annotations, achieving a 2.79 compression ratio on the OWL 2 Functional syntax (which is by itself the least verbose syntax). In this way, each tag must contain only 530 B on average, allowing compatibility with several off-the-shelf NFC products, e.g., the NXP NTAG216 (https://www.nxp.com/products/rfid-nfc/nfc-hf/ntag/ntag-for-tags-labels) endowed with 888 B of memory.

*Processing times*. Experiments have been carried out on a Samsung Galaxy S6 by performing the four tasks reported in Figure 10.

Initialize reasoner: the mobile reasoner is initialized and the reference knowledge graph is loaded from file and parsed.Retrieve annotation: a sample compressed annotation containing a complete patient profile is retrieved from an NFC tag (annotation length was 848 B to fill up the tag memory and evaluate the longest communication delay).Check therapy: the selected therapy is verified with respect to the patient’s profile via Concept Abduction and Concept Contraction.Suggest alternatives: the system provides a ranked list of appropriate therapies.

Each test was repeated five times and average values were taken. Reasoner initialization resulted as the slowest task. However, this is not a relevant issue, since it occurs only once at DSS start-up. On the contrary, both NFC-based communication and matchmaking tasks can be deemed as compatible with on-the-fly CDS, thanks to the optimized data structures and inference algorithms specifically developed for mobile architectures.

## 6. Conclusions and Future Work

This paper presented a novel clinical decision support framework, based on an evolution of NFC standard data exchange format to support knowledge representation languages. The proposed approach leverages annotated descriptions of medications to be administered and patient’s medical profile to assist healthcare practitioners with therapy management. An OWL 2 knowledge graph provides the background conceptualization on the medical domain, while semantic metadata are stored in NFC tags attached to pharmaceuticals packaging, patients’ wristbands, practitioners’ badges and medical equipment. Architecture and user interaction of the proposed DSS are oriented toward fully mobile and pervasive usage. It helps the domain expert in an unobtrusive way, by automatically invoking inference procedures upon relevant knowledge fragments extracted directly via NFC. Semantic matchmaking is exploited to detect therapy contraindications and interactions, as well as to suggest the best care options.

The pervasive CDS has been validated in a therapy verification case study in rheumatology, therefore the proposed KG currently focuses on a rather narrow set of diseases.This can be considered as the main limitation of the approach from a theoretical point of view. Nevertheless, scalability and the capability of being extendable to multiple applications is one of the main benefits of the proposed approach, as no centralized infrastructure is required and the described knowledge modeling patterns are inherently general.

Other weaknesses of the proposed framework, depending on the particular application scenario, are as in what follows:Integration with local and regional standards for electronic health records has not been studied yet. In combination with the limited storage space of NFC tags, it may require minimum-loss semantic summarization methods [67] in addition to lossless data compression.Further case studies have been envisioned but not developed yet. In particular, first aid, ambulance transport and tele-homecare scenarios provide additional challenges: due to this reason they are necessary for a full validation—and possibly a refinement—of the framework from architectural and technological viewpoints.The interoperability between the patient’s NFC tag and wearable health monitoring sensors is a particularly meaningful development, as the latter may be exploited to update the health status by means of semantic-based data mining and annotation [47] in a fully decentralized and automated fashion.Prospective clinical studies are needed to assess the effective impact of the proposal on healthcare quality. Differently from the conducted case study, they must involve a large number of healthcare professionals in multiple facilities for a long time span.

Future work aims to expand the scope of the knowledge graph for decision support toward further medical fields, to investigate novel HCI models exploiting e.g., Mobile Augmented Reality (MAR) [68], and to address all the above aspects.

## Figures and Tables

**Figure 1 sensors-20-04923-f001:**
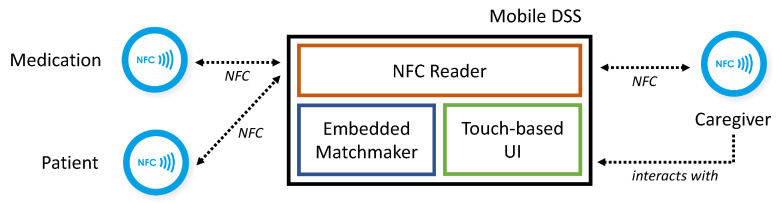
Clinical decision support system architecture.

**Figure 2 sensors-20-04923-f002:**
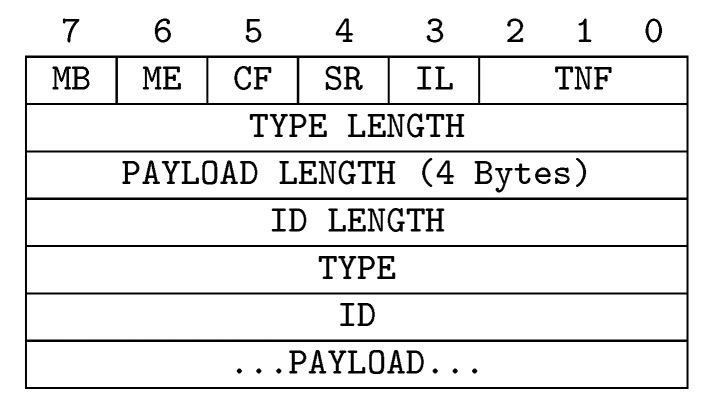
Structure of NDEF records.

**Figure 3 sensors-20-04923-f003:**
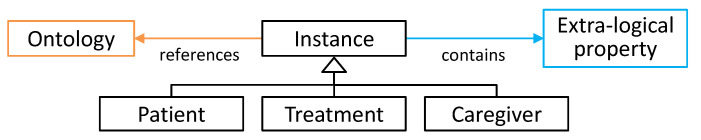
Knowledge model.

**Figure 4 sensors-20-04923-f004:**
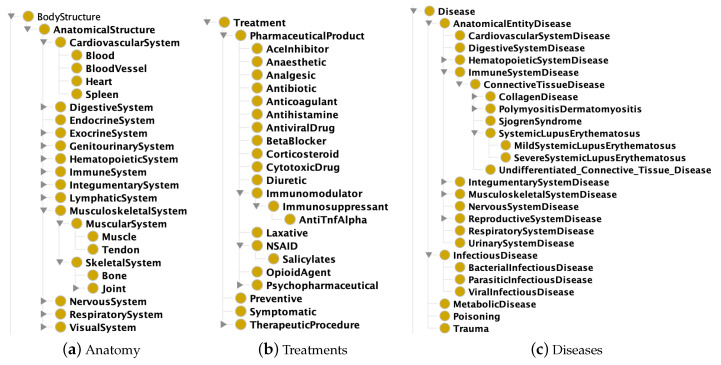
Conceptual model of the medical domain.

**Figure 5 sensors-20-04923-f005:**
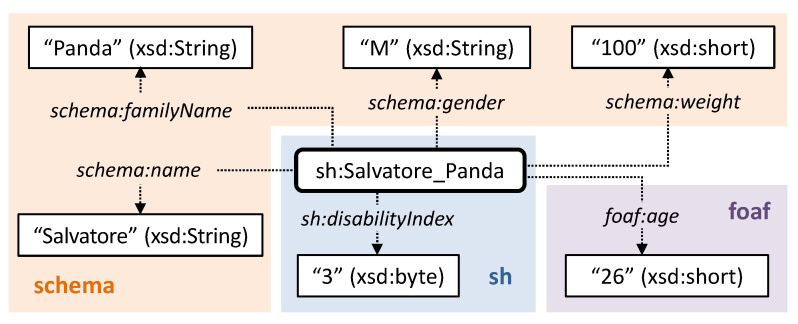
Example of context-aware attributes modeling for patients.

**Figure 6 sensors-20-04923-f006:**
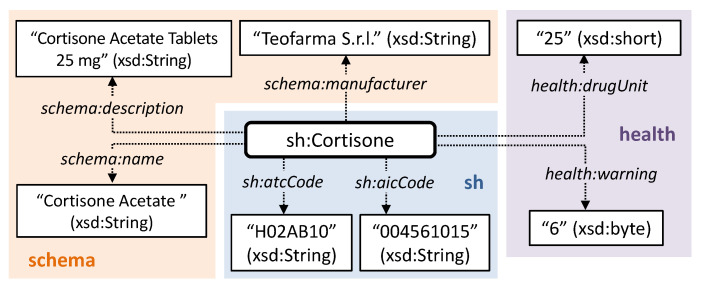
Example of context-aware attributes modeling for medications.

**Figure 7 sensors-20-04923-f007:**
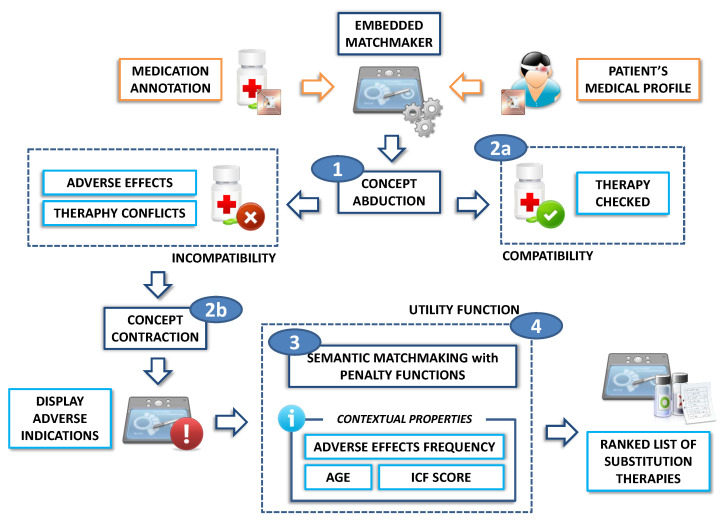
Reference framework for therapy verification.

**Figure 8 sensors-20-04923-f008:**
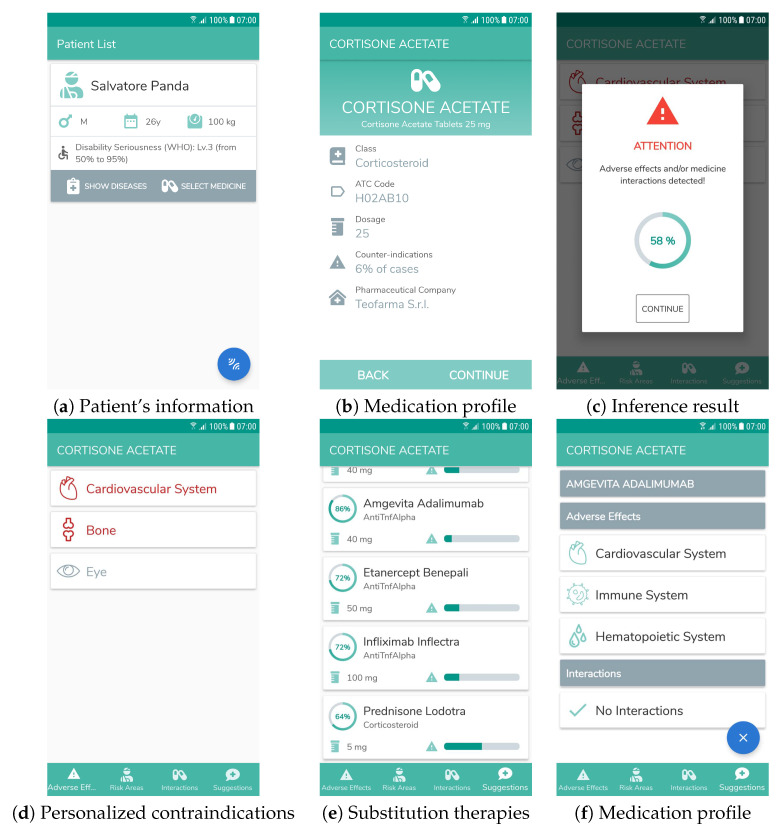
Screenshots of the mobile DSS user interface for Example 1.

**Figure 9 sensors-20-04923-f009:**
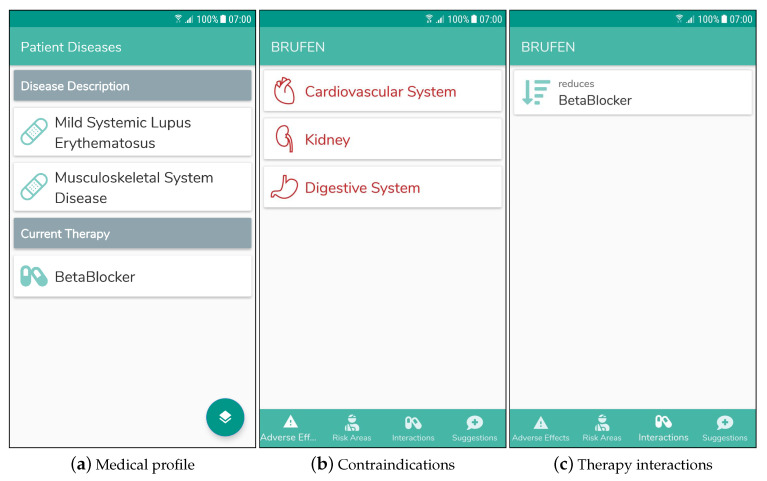
Screenshots of the mobile DSS user interface for Example 2.

**Figure 10 sensors-20-04923-f010:**
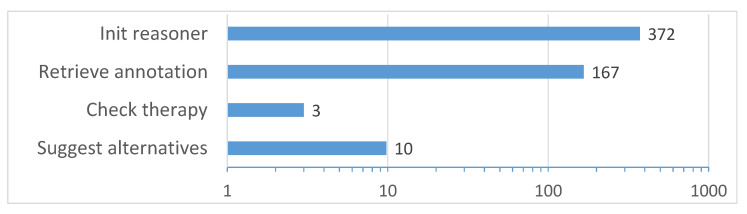
Processing time (ms).

**Table 1 sensors-20-04923-t001:** Comparison of the proposed approach with related Clinical Decision Support systems.

Work	Year	Main Use Cases	AutoID or Tracking Technology	Repre- Sentation Language	Decision Support Mechanism	Match Types	Resource Ranking	Main Contribution
[22]	2009	Patient location tracking, med- ication verifi- cation, inventory management.	RFID (simulated)	N/A	SQL-like query	Exact only	No	Early RFID- based HMS with rigorous design and test.
[41]	2010	Medication administration and verification.	EPCglobal RFID (simulated)	Semantic (DIG, in XML syntax)	Semantic match- making	Exact, approx- imated	Yes	Non-standard inferences for patient-medica- tion verification and discovery.
[23]	2017	Patient admis- sion, remote health mon- itoring, medica- tion prescription.	NFC	N/A	SQL-like query	Exact only	No	Integration of NFC and wire- less sensors.
[34]	2017	Patient admission, visit, treatment.	NFC	N/A	SQL-like query	Exact only	No	Multi-agent architecture for clinical process design.
[35]	2017	Ambient-assisted living.	NFC, general IoT sensors	Semantic (OWL, in JSON or XML syntax)	SPARQL query, OWL-S composition	Exact only	No	Virtual objects, semantic-based service composition.
[24]	2019	Medication prescription and administration.	NFC	N/A	SQL-like query	Exact only	No	Secure mutual patient-staff authentication.
[15]	2018	Medication administration and verification.	NFC	N/A	SQL-like query	Exact only	No	Rigorous de- sign and tests with nurses.
[25]	2020	Remote health monitoring.	General IoT sensors	Numeric	SVM + correlation + rules	Exact, approx- imated	No	Secure authen- tication, anom- aly detection, cope with missing data.
[43]	2018	Ambient-assisted living.	General IoT sensors	Semantic (OWL)	SPARQL query + SWRL rules	Exact only	No	Semantic-based personalized appliance discovery.
[48]	2019	Human Activity Recognition.	3D motion sensor (Kinect)	Semantic (OWL)	Semantic match- making	Exact, approx- imated	Yes	Posture/gesture recognition via semantic matchmaking.
[46]	2020	Human Activity Recognition.	General IoT sensors (simulated)	Numeric	Machine learning	Exact, approx- imated	No	Comparison of machine learn- ing techniques.
This paper	2020	Medication administration and verification.	NFC	Semantic (OWL, in various syntaxes)	Semantic match- making	Exact, approx- imated	Yes	Verification of medication interactions and contraindi- cations via non-standard inferences.

**Table 2 sensors-20-04923-t002:** NDEF Type Name Format values.

TNF	Name	Description
0	Empty	No payload data, typically used for newly formatted tags.
1	Well-known	Payload data format defined by the Record Type Definition (RTD) specification [54].
2	MIME	Payload data format specified through a MIME type.
3	URI	Reference to a resource identified by a generic Uniform Resource Identifier (URI).
4	External	User-defined data defined according to the RTD specification.
5	Unknown	Unknown data format, in this case the type length field is always zero.
6	Unchanged	Used for chunked record, the specific TNF is defined in the first record of the chunked set.
7	Reserved	Reserved for future use

**Table 3 sensors-20-04923-t003:** Average size of compressed annotations.

Compression	OWL/XML	OWL/RDF	Functional	Manchester	Turtle
Plain Text	3915.00	6388.74	1480.74	1968.00	4984.95
EXI	668.11	645.63	–	–	–
HDT	–	–	–	–	3534.42
LZMA	814.68	994.05	568.32	641.26	880.42
GZIP	796.42	996.89	542.32	619.11	869.32
Deflate	784.42	984.89	**530.32**	607.11	857.32

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
