# Peer review of "Knowledge-Based Decision Support in Healthcare via Near Field Communication"

_sensors, 2020, doi:10.3390/s20174923_

Round 1

Reviewer 1 Report

In this article, the authors have proposed a DSS system using NFC for HealthCare system. Even though the framework and messages are pretty standard ones. However, novelty of the proposed approach is still my concern.  The phone is a bit old phone...

There are some other issues such as literature review is weak. Many important papers are missing.

Please compare your DSS system with respect to the following

https://ieeexplore.ieee.org/abstract/document/9134756

https://www.sciencedirect.com/science/article/pii/B9780128190432000058

Author Response

Reviewer 1

Q 1.1 The phone is a bit old phone...

Response

Thanks for the observation, which allows us to clarify that the choice of Galaxy S6 (a 5-year-old phone model) and Nexus 4 (8 years old) had the purpose of assessing whether our approach could achieve acceptable performance even on relatively resource-constrained hardware. We have clarified this point in our revised manuscript. The Nexus 4 was used only to emulate NFC tags via the Android Beam API, while all computational tests have been run on the Galaxy S6, as explained in the manuscript.

Q1.2 There are some other issues such as literature review is weak. Many important papers are missing. Please compare your DSS system with respect to the following https://ieeexplore.ieee.org/abstract/document/9134756 https://www.sciencedirect.com/science/article/pii/B9780128190432000058

Response

We are very grateful to the reviewer for the comment. We have included a comparison of our framework with further relevant works in Section 2, including the suggested ones.

Reviewer 2 Report

Based on an evolution of NFC standard data exchange format to support knowledge representation languages, a decision support framework for healthcare is presented, in which, annotated descriptions of medications and patient’s case history are stored in NFC transponders and used to help caregivers providing the right therapy.

However, there are still some suggestions.

  1. Remove all typos and other grammatical errors. For example, “in a treatment, suggesting substitute therapies” should be “in treatment, suggesting substitute therapies”. “practictioner” should be “practitioner”.
  2. On the evaluation of proposed knowledge-based decision support, the authors should select the latest research achievements for comparison studies to enhance the convincing for the proposed scheme.

Author Response

Reviewer 2

Q 2.1 Remove all typos and other grammatical errors. For example, “in a treatment, suggesting substitute therapies” should be “in treatment, suggesting substitute therapies”. ‘practictioner” should be “practitioner”.

Response

Many thanks for the remarks. We have fully revised our manuscript for removing typos and language errors.

Q 2.2 On the evaluation of proposed knowledge-based decision support, the authors should select the latest research achievements for comparison studies to enhance the convincing for the proposed scheme.

Response

We are grateful for the suggestion. We have added comparisons with novel and recent relevant works to Section 2.

Reviewer 3 Report

The work presents an innovative framework DSS based on the NFC standard. The proposed approach leverages annotated descriptions of medications to be administered and patient’s medical profile to assist healthcare practitioners with therapy management. It has been applied in a case study but it seems that it can be expanded easily.

Author Response

Reviewer 3

Q 3.1 The work presents an innovative framework DSS based on the NFC standard. The proposed approach leverages annotated descriptions of medications to be administered and patient’s medical profile to assist healthcare practitioners with therapy management. It has been applied in a case study but it seems that it can be expanded easily.

Response

Many thanks for the comment. Really the scalability and the capability of being extendable to multiple applications is probably the main benefit of the proposed system. This has been better pointed out in the revised manuscript.

Round 2

Reviewer 1 Report

Thank you for revising the paper. However, I think still some works to be done.

  1. In Section 2, include a table to show what are the issues in the existing works. Why you are proposing a new model?

    2. Motivation part needs to clearly define , why we need NFC?

    3. NFC-based system also suffers from some Challenges , you need to mention those.

    4. What is the weakness of you module? 

Author Response

Thank you again to the reviewer for the effort spent in promptly revising our work and in particular for the provided feedback and suggestions. Changes with respect to previous revision are in red in the paper. Detailed replies to remarks are reported in what follows.

--Reviewer 1--

-Q 1.1 In Section 2, include a table to show what are the issues in the existing works. Why you are proposing a new model?

-Response
We are grateful for this suggestion. We have included a table at the end of related work section, reporting limitations of existing approaches our proposal is able to improve upon. For each work, we also highlight what we consider as the main novel contribution it has brought to the state of the art.

-Q 1.2 Motivation part needs to clearly define, why we need NFC?

-Response
We have included a clearer motivation for the adoption of NFC in the Introduction, providing more specific considerations and coalescing the ones which were scattered in other parts of the paper.

-Q 1.3 NFC-based system also suffers from some Challenges, you need to mention those.

-Response
Thank you for this advice. We have added considerations about the challenges of NFC in the Introduction, next to the new motivation paragraph.

-Q 1.4 What is the weakness of you module?

-Response
Many thanks for the inquiry. We have clarified limitations and weaknesses of our proposal in the Conclusion of the revised manuscript.